# NDN Content Store and Caching Policies: Performance Evaluation

Elídio Tomás da Silva [1,2,*], Joaquim Melo Henriques de Macedo [2] and António Luís Duarte Costa [2]

1   Faculty of Engineering, Lurio University, Pemba 958, Mozambique
2   ALGORITMI Centre, School of Engineering, University of Minho, Campus Gualtar, 4710-057 Braga, Portugal; macedo@di.uminho.pt (J.M.H.d.M.); costa@di.uminho.pt (A.L.D.C.)
*   Correspondence: id6644@alunos.uminho.pt

**Abstract:** Among various factors contributing to performance of named data networking (NDN), the organization of caching is a key factor and has benefited from intense studies by the networking research community. The performed studies aimed at (1) finding the best strategy to adopt for content caching; (2) specifying the best location, and number of content stores (CS) in the network; and (3) defining the best cache replacement policy. Accessing and comparing the performance of the proposed solutions is as essential as the development of the proposals themselves. The present work aims at evaluating and comparing the behavior of four caching policies (i.e., random, least recently used (LRU), least frequently used (LFU), and first in first out (FIFO)) applied to NDN. Several network scenarios are used for simulation (2 topologies, varying the percentage of nodes of the content stores (5–100), 1 and 10 producers, 32 and 41 consumers). Five metrics are considered for the performance evaluation: cache hit ratio (CHR), network traffic, retrieval delay, interest retransmissions, and the number of upstream hops. The content request follows the Zipf–Mandelbrot distribution (with skewness factor $\alpha = 1.1$ and $\alpha = 0.75$). LFU presents better performance in all considered metrics, except on the NDN testbed, with 41 consumers, 1 producer and a content request rate of 100 packets/s. For the level of content store from 50% to 100%, LRU presents a notably higher performance. Although the network behavior is similar for both skewness factors, when $\alpha = 0.75$, the CHR is significantly reduced, as expected.

**Keywords:** caching replacement policies; named data networking; performance evaluation



## 1. Introduction

Named data networking (NDN) [1,2] is a content-centric and name-based network architecture. The content identification is based on names, which must be global in the case of a global network. Identifying content through names provides their dissociation from their location (addresses). This characteristically results in the advantage of being able to store (cache) and easily share content among several consumers.

The caching performance in NDN depends on a combination of several factors, such as (a) content routing; (b) decision for content placement (caching decision strategy); (c) decision for content replacement (caching replacement policy) [3,4]; and (d) the network topology. The networking research community is working on the development of these topics.

The present work aims at evaluating the performance of caching replacement policies in a network topology with a variable number of *content stores* (CS). Specifically, the performance of *least recently used* (LRU), *least frequently used* (LFU), *first in first out* (FIFO), and random policy are evaluated. Seven network scenarios are taken into consideration. Each scenario corresponds to a network, where the CS are deployed on a varying number of routers corresponding to 5%, 20%, 30%, 40%, 50%, 80% and 100% of the total nodes.

Although this work does not perform an exhaustive survey on similar studies, some representative proposals can be found in [5,6] for performance evaluation on a caching system with a variable number of CS, or in [7–16] for evaluation with a variable size of a

fixed number of CS, as presented in Section 4. A survey on replacement schemes, which can complement this Section, is performed in [17]. However, none of the aforementioned studies evaluate and compare the four replacement policies considering different scenarios, or present all the commonly used metrics for performance evaluation. The used metrics are cache hit ratio (CHR), retrieval delay, number of upstream hops to cached content, number of re-transmissions on cache miss, and the network traffic in terms of number of transmitted messages.

A comprehensive evaluation of the existing caching replacement policies and caching decision strategies is necessary to better decide on the adequate combination of these components for a specific network. The evaluation is also important because its results can likely expose possible issues still to be solved. The present work only focuses on the main four caching replacement policies and only one caching decision strategy. Future work may extend the present study to include other existing policies, such as the ones presented in Section 3.2, and combine them with different caching decision strategies, such as the ones presented in Section 3.1.

Our main contributions can be summarized as in (1) complementing the existing studies that evaluate the performance of caching policies in CCN/NDN; (2) extending the analysis to cover more caching policies simultaneously, with relatively much more traffic per node (100 packets/s, in contrast with 5 packets/s from previous studies); (3) highlighting where discordant results exists from previous studies; and (4) differently from existing studies, we additionally evaluate the caching performance under a relatively complex network composed of a higher number of content sources (producers).

The contributions of this work are decisive on advising a better caching system configuration for a specific network in terms of (a) the metrics to consider; (b) the percentage of caching nodes in the network; (c) the adequate caching policy to apply; and (d) the complexity of the network in terms of traffic and varying number of content sources.

The remainder of this work is organized as follows: Section 2 presents the overall structure of the NDN architecture. Section 3 presents the in-network caching in NDN. Section 4 presents related work. In Section 5, the simulation environment is described. The results of the simulations and respective discussion are also presented in this section. The conclusion is presented in Section 6.

## 2. The NDN Architecture Overview

The IP-based internet is a host-centric network in which the communication between two hosts demands previous knowledge of the respective addresses in the network.

Since the standardization of this protocol stack, the way communications take place has changed profoundly. Current communications are content-centric based, i.e., the consumer is interested in the content and requests it by its name, instead of requesting it by its location (address).

With the rapid growth in electronic commerce (e.g., Amazon), live multimedia (video in real time, e.g., Netflix) social networks (e.g., Facebook and WhatsApp), new challenges for the TCP/IP protocol stack have emerged, demanding the development of new architectural paradigms for this new reality, i.e., content-centric solutions.

Specifically, the need to solve issues, such as the node mobility, network security, data privacy, network scalability, and multi-path routing, which remain challenging issues for TCP/IP-based internet, have forced the emergence of several research projects, e.g., the *content-centric networks* (CCN) [18], *network information* (NetInf) [19], *data-oriented network architecture* (DONA) [20], and *publish–subscribe internet routing paradigm* (PSIRP) [21], where NDN, a particular implementation of CCN, is considered the most promising solution [2,22]. The architectural principles which guided the design of NDN are specified in [1,23].

NDN adopts a pull-based communication model, where a consumer initiates the communication process by sending a request for the desired content by means of an interest. There are two types of packets, *interest* and *data* packets.

When an interest is sent out, it is forwarded (upstream path) through different nodes toward the content source. When the corresponding content is located, the data are sent back (downstream path) to the requester by breadcrumbs. That is, the data packet is not forwarded. Instead, the status information created and saved along the upstream is used to guide the data back. One of the main issues eliminated with this approach is packet (data) looping. Interest looping is avoided by combining the *content name* (CN) with a *nonce*—a random number generated to unambiguously identify an interest.

Each NDN node contains a structure composed by the following three elements for packet (interest and data) management: (1) *content store* (CS); (2) *pending information table* (PIT); and (3) *forwarding information base* (FIB).

The CS is used to increase the probability of sharing content, saving bandwidth and decreasing the response time for content request. The cached copy is forwarded to any consumer who requests it, without the need to make the request again to the content source. More details on CS are presented in Section 3.

The PIT stores information about the pending interest, and a recently served interest. If an interest reaches a given router and misses its corresponding content, it is forwarded to other nodes. Before the interest is forwarded, its CN and ingress face are registered on PIT. This information is then used to send the data back, downstream.

The life of an interest in PIT is generally equal to the *round trip time* (RTT). The pending interest can be removed from PIT even before RTT elapses. In this case, the request is not satisfied, and it is the responsibility of the consumer to re-issue the request [1,24].

FIB can be populated by a name–prefix-based routing protocol, and it is where the routing information is stored. The incoming interest that misses the respective contents in a specific router cache, after having been added to PIT, is forwarded to the next hop based on the information provided partially by FIB. The existence of more than one outgoing face registered on a specific FIB entry for a given forwarded interest indicates that this interest was forwarded through more than one path. If none of the FIB entries matches the interest, then it is forwarded through all of the node's faces.

In the case of failure in forwarding an interest due to path congestion, insufficient information in FIB about the prefix, or any other error, a *negative acknowledgment* (NACK) is sent back (downstream) with information indicating the possible cause of failure.

## 3. In-Network Caching

Caching in NDN is similar to the buffer's caching concept in TCP/IP. However, since this architecture is based on end-to-end communication and the identification of content is associated with its location (address), packets stored in the buffer are useless as soon as they are sent to their destination. NDN caching is more granular, ubiquitous and transparent to applications, unlike what happens, for instance, with *content delivery networks* (CDN) [25] and web caching.

Caching benefits the network in such aspects as (a) the higher probability of sharing content, saving bandwidth and decreasing the requesting delay; (b) the dissociation of the content from its producers, improving mobility for both consumers and producers; (c) the reduction in latency in content dissemination; and (d) the elimination of a single point of failure.

Although it is desirable to keep the content cached for a long period to allow continuous sharing, the size of the CS is a limiting factor. Several studies have been performed in order to determine the best way to extend this period, or, when the replacement is unavoidable, ensure that it does not negatively affect the whole network's performance.

Caching performance in CCN depends on factors such as (a) the content routing; (b) the decision for content placement (caching strategy); and (c) the decision for content replacement (caching policy) [3,4]. It is usually measured based on metrics such as (a) the cache hit ratio; (b) the network traffic; (c) the average number of upstream hops; and (d) the delay for data delivery.

Content forwarding is an intensively investigated area in NDN. Some reviews in this area can be found in [26,27]. Some routing studies with particular attention on caching issues are presented in [4].

### 3.1. Caching Decision Strategy

The distribution of a cache along the network is decided by the caching decision strategy. Several strategies have been proposed. The main ones are (a) *leave copy everywhere* (LCE)—when a data packet is sent downstream, a copy of it is cached in all nodes along the path; (b) *leave copy probabilistically* (LCP)—similar to LCE but the content is cached with a given probability; (c) *leave copy down* (LCD) [28]—when a content is sent downstream, the cached copy is replicated one hop downstream. This scheme avoids the proliferation of the same content throughout the network; (d) *move copy down* (MCD) [28]—similar to LCD but, in this case, the strategy *shifts* the cached copy one hop downstream; (e) *copy with probability* (*Prob*) [28]—a copy is cached in all downstream nodes but with a probability $p$. For $p = 1$, this strategy is equal to LCE; (f) *randomly copy one* (*RCOne*) [29]—similar to LCD but instead of caching in the next hop node, the copy is placed on a randomly selected downstream node; (g) *probabilistic cache* (*ProbCache*) [30]—caches in the downstream nodes, but with a different probability for each node. The probability of placing a copy in a node closer to the consumer is higher than that far from the consumer; and (h) *WAVE* [31]—similar to LCD but this scheme takes into account the correlation in requests between the different chunks from the same content. A node explicitly suggests to the next downstream node which content should be stored. In this work, as we do not focus on the caching decision, only the LCD strategy is considered. An extension of this work to include other strategies may be considered in the future.

The content popularity is another factor contributing to caching performance. Some proposals on categorizing the content popularity can be found in [4]: (a) popularity over web proxy [32]; (b) popularity over hierarchy of proxies [33,34]; (c) popularity in *peer-to-peer* (P2P) networks [35]; (d) popularity of *video on demand* (VoD) [36]; and (e) popularity of user-generated content (video on Youtube) [37–39]. According to these studies, the *Zipf–Mandelbrot* (MZipf) distribution [32] is the most suitable model to represent the content popularity for web requests. The probability of accessing an object at rank $i$ out of $N$ is given by (1)

$$p(i) = \frac{K}{(i + q)^\alpha} \tag{1}$$

where $K$ is given by

$$K = \sum_{c=1}^{N} \frac{1}{(c + q)^\alpha}. \tag{2}$$

The skewness factor ($\alpha$) is used to control the slope of the distribution curve, and $q \geq 0$ defines the flatness of the curve. The higher the skewness factor $\alpha$, the higher the CHR. We chose two values for $\alpha$: one observed in traces for web proxies ($\alpha = 1.1$) which mimic an ISP network, and its content distribution follows the heavy-tailed distribution [33]; the other for user-generated web content ($\alpha = 0.75$) [32]. The reasoning behind this choice is the need to study the influence of different CS levels for different content types.

Within the scope of strategies based on popularity, some additional proposals can be found in such works as [40–46].

### 3.2. Caching Replacement Policies

The size of the CS is unavoidably limited. When it is necessary to cache new content and the CS is full, some content must be replaced. The caching replacement policy decides on what content to replace. When the requested content is found in CS, a *cache hit* occurs; if not, it is a *cache miss*. The cache miss occurs when the content is never forwarded to the node or because, having already been forwarded, it is eventually replaced. Several caching policies have been proposed, the main ones being (a) *least recently used* (LRU)—based on

access time. This policy searches and replaces the content not requested any longer; (b) *most recently used* (MRU)—also based on access time, as in the case of LRU. In this case, the most recently requested content is replaced; (c) *least frequently used* (LFU)—based on time and frequency of access/request. This policy searches and replaces the less often requested content. Unlike LRU or MRU, which only use time to evaluate what content to replace, this policy uses two factors, age and the access frequency; (d) *first in first out* (FIFO)—based on time, similar to LRU, but here, the order of content arrival is taken into account, that is, differently from LRU, the new requests, finding an already stored and least recently used object, do not *refresh* the timer associated to it [8]; (e) *priority first in first out* (*Priority-FIFO*) [47]—based on time, similar to FIFO but with three queues of relative priorities (the unsolicited queue, the stale queue, and the FIFO queue); and (f) *random*—in this policy, the content to replace is randomly selected.

Some extensions of these main policies can be found in reviews, such as [8,48]. Specific reviews on NDN caching can be found in studies, such as [4,49].

## 4. Related Work

As the performance of the caching system is a very important factor to support the content-centric networks, and one that can affect the entire network performance, it has been the focus of attention by the networking research community. Not intending to be exhaustive, the following are some of the related works found in this area.

In [50,51], a caching replacement policy dubbed *universal caching* (UC) was proposed. The proposal was evaluated and compared to FIFO and LRU. The authors concluded that the proposal presents better performance, even in situations where the CS size and topology are variable (location of consumers and producers in the network). The study was performed using two scenarios: (1) a synthetic topology based on the *Watts-Strogatz* model [52], more suitable for networks with high density but short range, as in the case of sensor networks; and (2) *spring* topology [53], more suitable for the *internet service provider* (ISP). In [5], the influence of number of CS on the performance of the network was investigated. The study used two network topologies suitable for ISP (e.g., Abilene [54] with 11 nodes, and GEANT with 42 nodes) and compared the performance of LFU in a topology with 20%, 50%, 80% and 100% of CS. The study concluded that the network performs better when the network has around 40% of CS; above this threshold, the performance drops. The study [7] proposed *content caching popularity* (CCP), a replacement policy based on content popularity. The authors based their study on a network where the consumers and producer are placed on the edge of the network. They evaluated and compared the performance of LRU and LFU in terms of CHR, server load, and average network transfer rate. The work in [55] evaluated the gain of caching strategies, comparing the placement of CS at the edge or distributed pervasively. The authors concluded that the difference in these two approaches is relatively low, around 9%, which significantly reduces to around 6% when increasing the size of edge caching. In [6], *networking cache-SDN* (NC-SDN), a centralized caching replacement strategy based on *software-defined networks* (SDN) [56] and popularity was proposed. The popularity is computed by switches, based on the number of occurrences of a certain prefix in FIB. The study performed a comparative study with LFU, LRU, FIFO and NDNS [57], and concluded that NC-SDN is better than the others in terms of CHR and bandwidth occupation. The authors in [13] proposed *PopuL*, a cache replacement policy based on content popularity and the betweenness of nodes in the network. An intra-domain resource adaptation resolving server was used to store the cache status. Using CHR metric, the proposal was evaluated and compared with LRU, LFU and FIFO. The authors concluded that *PopuL* presents better performance. The authors in [12] proposed *PopNetCod*, a popularity-based caching policy that decides on caching or evicting a content based on the received interest packet instead of the corresponding data packet, that is, the decision of caching is made while processing the received interest before the corresponding data packet arrives at the router. *PopNetCod* includes a caching decision strategy and a cache replacement policy. The proposal was evaluated and the

cache replacement policy was compared with LRU. The authors concluded that *PopNetCod* is better than the others in terms of the chosen metrics. A popularity-based cache policy for privacy-preserving in NDN (PPNDN) was proposed by [10]. The solution is designed to prevent cache privacy violation, resorting to the cache expiration time, which is set according to the number of content requests. The cache probability of the content in PPNDN is calculated based on the content popularity and the router information. The proposal is evaluated (based on the CHR, cache storage time, and average load of content provider), and its performance is compared with LRU and FIFO. The authors concluded that PPNDN is better than the others in terms of the chosen metrics. Management mechanism through cache replacement (M2CRP), a caching replacement policy based on multiple metrics (i.e., the content popularity; the freshness; and the distance between the location of two specific nodes) was proposed in [11]. When a replacement was required, the proposal used the aforementioned factors to calculate the candidacy score for each content in CS. The average weight of each of these parameters was computed, and the content with the lower weight was the candidate for replacement. The proposal was evaluated and compared with LRU and FIFO, in terms of CHR, interest satisfaction delay and interest satisfaction ratio. The authors concluded that the proposed solution presents better performance than others in terms of the chosen metrics. The immature used (IMU) replacement policy was proposed in [15]. Besides the popularity, the proposal resorts to the content maturity index, which is calculated by using the content arrival time and its frequency within a specific time frame. When CS is full, the least immature content is evicted. The performance evaluation based on CHR, path stretch, latency, and link load is performed and compared with FIFO, LRU, LFU and a combination of LRU and LFU (LFRU) [58]. An adaptive least recently frequently used (LRFU) [59] replacement policy was proposed by [14]. The proposed solution calculates the content distribution by its recency and frequency, and is capable of adjusting the heap list and linked list in the router CS based on the CS size. The proposed solution is evaluated in terms of network latency, delay distribution, CHR, and miss rate, and compared with LFU, LRU and LRFU. The authors concluded that the solution presents better performance than the others in terms of the chosen metrics. In order to reduce the cost of estimating content popularity, [16] categorized data for caching (CaDaCa) was proposed, which is based on data categorization (e.g., politics, sports, and science), to make the content names more informative. The proposal includes a caching decision strategy and a cache replacement policy. The proposal was evaluated (based on CHR, hop reduction ration, and server load ratio metrics) and the cache replacement policy was compared with LRU and FIFO.

Table 1 shows and summarizes related work in terms of their specific configuration for simulation.

**Table 1.** Some studies on caching policies. All studies use LCD as the default caching strategy.

| Reference | Topology (# Nodes) | Caching Policy | Number of CS, in % | Size of CS |
|---|---|---|---|---|
| Aubry et al. [5] | Abilene (11), GEANT (41) | LFU | 20, 40, 50, 80, 100 (of # nodes) | 1000 contents |
| Ran et al. [7] | Hierarchical (20) | LFU, LRU, *CCP* [2] | 100 (of # nodes) | 40–60% of 200 contents |
| Shailendra et al. [51] | Synthetic, *Sprinter* (52) | FIFO, LRU, *UC* [2] | 100 (of # nodes) | (2–20)% of cache [1] |
| Panigrahi et al. [50] | Synthetic, *Sprinter* (52) | FIFO, LRU, *UC* [2] | 100 (of # nodes) | (2–20)% of cache [1] |
| Kalghoum et al. [6] | Hierarchical (150–3000) | LRU, LFU, FIFO, *NC-SDN* [2] | 5 (of # nodes) | [1] |
| Yang and Choi [10] | Synthetic (10) | LRU, FIFO, *PPNDN* [2] | 30 (of # nodes) | 10–70% of 1000 contents |
| Ostrovskaya et al. [11] | Vehicular network (108) | LRU, FIFO, *M2CRP* [2] | 100 (of # nodes) | 50, 75, 100, 125, 150 MB |
| Liu et al. [13] | Binary tree topology (12) | LFU, LRU, FIFO, *PopuL* [2] | 100 (of # nodes) | (10–190) MB |
| Saltarin et al. [12] | Layered topology (169) | LRU, *PopNetCod* [2] | 27 (of # nodes) | 0.9–2.3% of 540,500 contents |
| Putra et al. [14] | Substrate topology (15) | LRU, LFU, LRFU, *adaptive LRFU* [2] | 100 (of # nodes) | 10–100 blocks |
| Rashid et al. [15] | GEANT (100) | LRU, FIFO, LFU, LFRU, *IMU* [2] | 100 (of # nodes) | 4–20% of 100,000 contents |
| Abdelkader Tayeb et al. [16] | *k-ary* trees ([1]), Scale free topology ([1]) | FIFO, LRU, *CaDaCa* [2] | 100 (of # nodes) | 10% of 1000 contents |
| Present work | Testbed (43), Abilene (11) | LFU, LRU, FIFO, Random | 5, 20, 30, 40, 50, 80, 100 (of # nodes) | 1000 contents |

[1] Value not provided. [2] Policies proposed by the referenced authors.

In summary, the main differences, and thus the main contributions of the present work compared to the existing similar studies are as follows. The work by [5] only evaluated the performance of the LFU strategy, with the consumer request rate of 5 packets/s. The study did not consider the re-transmission and the hop-count metrics. As shown in the next

section, the conclusions from this study are partially divergent with the results achieved in the present work. The study by [7] evaluated the performance of LRU, LFU and their proposed replacement strategy considering the CHR, traffic and server load. The study did not indicate the consumer request rate. In addition, the number of CS in the network is fixed, only varying in size, which is defined as the percentage of cached capacity by the total capacity. Their results, with LRU performing better than LFU, is only consistent with one of our scenario composed by various consumers and only one producer. The work by [50,51] evaluated FIFO, LRU and their proposed replacement strategy. The metrics considered are the CHR and the hop-count. The study considered all nodes equipped with CS, only varying their size (2% to 20% of CS) based on the similarity of requests. Consumers request content at a rate of 1000 packets/s. All the other selected studies ([10–16]) present a fixed number of CS, only varying the size of the deployed CS, and compare the performance of their proposed solution with some of the state-of-the-art methods, as presented in the table.

All studies consider a simple network with only one content source.

## 5. Simulation and Discussion

This section presents the environment, results of the simulations and respective discussion.

### 5.1. Simulation Environment Setup

We use ndnSIM [60] for the simulation. ndnSIM is an open-source network simulator for the NDN architecture. It is based on and leverages the *network simulator* (NS-3) [61] specifically to (a) create the simulation topology and specify the respective parameters; (b) simulate the link layer protocol model; (c) simulate the communication between the different NDN nodes; and (d) record the simulation events. This simulator is implemented to simulate the NDN network layer protocol model, and can work on top of any link layer protocol model [62].

Two topologies are considered: (1) Topology 1—generic, using the NDN testbed [63] (as shown in Figure 1a) composed by 42 routers. Connected to each of these routers, there is just one leaf where the consumer (or producer—the only producer in the network) is installed on; and (2) Topology 2—more suitable for ISP, using the Abilene network (see Figure 1b) composed by 11 routers. Connected to each of these routers, there are 4 or 5 leaves where the consumers are installed. One of the 11 routers is reserved, and the unique producer in the network is installed on a leaf connected to this router. For both topologies, the simulation is performed with the consumer request rate fixed to (a) 5 packets per second, and (b) 100 packets per second. To further evaluate the performance with a relatively more complex network, both topologies were additionally configured with multiple producers (i.e., 10 producers and 32 consumers requesting at a rate of 100 packets per second). Furthermore, the consumer request rate was incremented to 1000 packets per second for Topology 1, with all routers equipped with CS. A generic network and another network more suitable for ISP are chosen. Combining these topologies with different skewness factors, from MZipf, one observed in ISP networks and other for general user-generated content, will provide a more general view of the performance of the network, with the varying number of the CS.

Seven different scenarios are prepared for both topologies. The scenarios are configured to have the routers equipped with CS at the following levels: 100%, 80%, 50%, 40%, 30%, 20% or 5% of the total routers. In scenarios where the percentage of nodes equipped with content stores is less than 100%, the caching nodes are randomly selected, but the positions of the consumers and the producer(s) are fixed. Nodes without CS cannot cache contents but can forward packets.

The simulation environment was configured with the parameters shown in Table 2. The producer provides a universe of $10^6$ different contents. The size of each CS is 1000 chunks. Consumers request content at a rate of 100 packets/s and 5 packets/s in both topologies, following the *MZipf* popularity model with a modeling factor $\alpha = 0.75$ and $\alpha = 1.1$. This is

done to simulate a scenario with user-generated content and a scenario of an ISP network, respectively. The interests are sent at constant average rates with a randomized time gap between two consecutive interests.

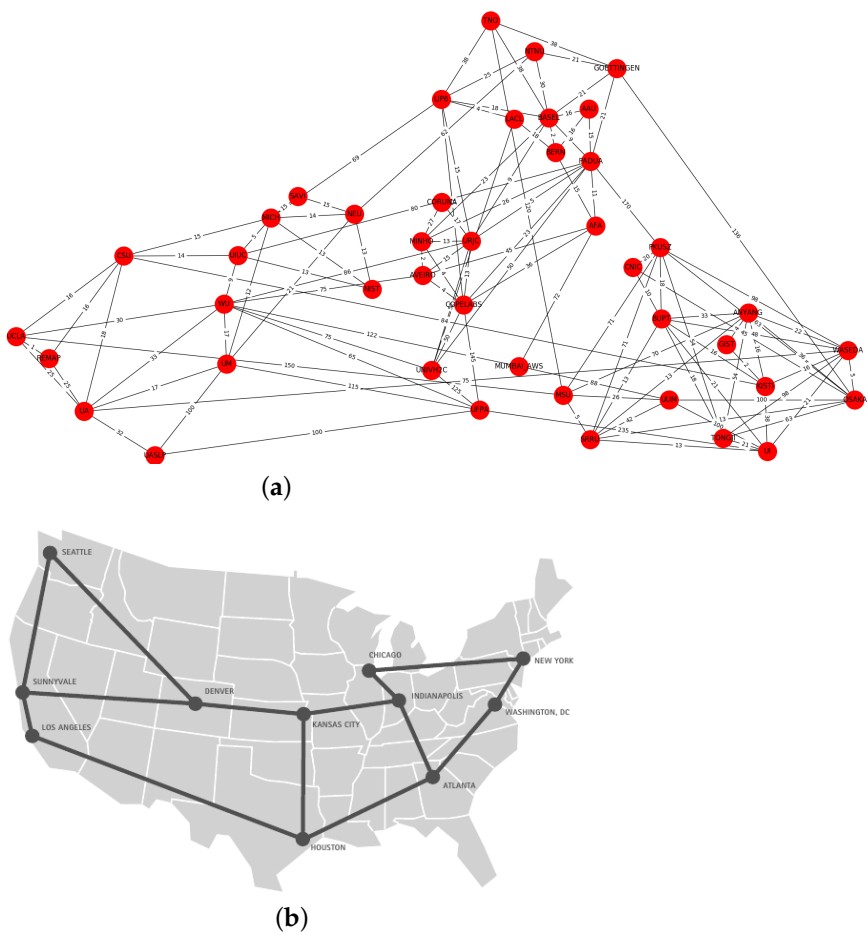

(**a**)

(**b**)

**Figure 1.** Used network topologies (NDN testbed and Abilene). (**a**) The NDN testbed topology [63], 43 nodes. (**b**) The Abilene network topology [54], 11 nodes.

**Table 2.** Parameters for the simulations.

| Parameters | Value |
| --- | --- |
| Topology (nodes) | NDN Testbed (42), Abilene (11) |
| Number of the catalogues | 1 |
| Size of the catalogue | $10^6$ different content |
| Number of producers | 1 (10) |
| Number of consumers | 41 (32) |
| Size of CS | 1000 contents |
| Popularity ratio | $MZipf$ ($\alpha = 1.1, \beta = 0.7$) and ($\alpha = 0.75, \beta = 0.7$) |
| Content request rate | 100 packets/s, 5 packets/s ($10^3$ packets/s) |
| Caching strategy | LCD |
| caching policy | LRU, LFU, FIFO, Random |
| Forwarding strategy | *Best Route* |
| Cache scenarios | 5%, 20%, 30%, 40%, 50%, 80%, 100% |
| Metric | Cache Hit ratio, Traffic, Delay, Re-transmissions, # hops |
| Simulations per scenario | 20 |
| Duration of simulations | 240 s, 600 s |

The best route forwarding strategy was used, and LCD was selected as the default caching strategy. As explained earlier, we only consider LCD given our interest in studying

only the replacement strategy in this work. For each scenario, 20 simulations are performed with the duration of 240 s each. Five metrics are considered for the performance evaluation: (1) the CHR, given by (3), which measures the performance of the system for a given caching policy; (2) the number of messages (traffic) in the network; (3) the delay between the time that a request is issued and the time (including time for re-transmissions) that the corresponding content is received on the requester; (4) the number of re-transmissions; and (5) the number of upstream hops.

$$Cache\ Hit\ ratio = \frac{\sum_{n=1}^{N} hit_j}{\sum_{n=1}^{N}(hit_j + miss_j)} \tag{3}$$

*5.2. Simulations Results*

This section presents the results of all performed simulations, considering the performance based on the aforementioned metrics.

5.2.1. Cache Hit Ratio

Figure 2 shows the network performance in terms of CHR, for the aforementioned two topologies and two different content request rates. In general, from this figure, we can conclude the following:

- The network performance (in terms of CHR) increases with the increased number of CS. This result is achieved because with more CS in the network, there is a higher probability of finding the requested content on CS along the route before reaching the content producer. When the number of CS is decreased, the available CS are quickly filled up, and the cache replacement occurs more frequently. It is important to note that the size of the CS plays a role in the magnitude of the results achieved here. For a reduced size, as is the case in this work, more frequently, the CS is filled up, requiring frequent replacement for a new content. The reduced CS size is deliberately chosen. Considering the defined consumer request rate, the resulting traffic is busy enough to fill up the CS and, thus, activate the replacement policy.
- Although for each replacement policy, the higher gain on using CS is noticeable when increasing the percentage of CS from 0 to 50%, the difference among the policies in terms of their performance is more noticeable with a higher number of CS in the network. This result is a good metric for advising a specific scheme for a given network.

    Specifically,

- Figure 2a shows LFU performing better with CS below 40%. After this threshold, LRU presents better performance. For CS above 50%, FIFO also presents a better performance than LFU and random.
- Figure 2b–d shows a similar, linear behavior, where, different from Figure 2a, LFU presents a better performance than all other policies. In addition, we can see that random performs now relatively well, with performance levels similar to LRU for a CS level above 50%.

Concerning the difference in terms of performance mainly between LFU and LRU, we recall that, different from LRU, which is only based on the access time to decide on eviction, LFU is based on both the time and frequency of the access/request and, therefore, performs well for more popular contents.

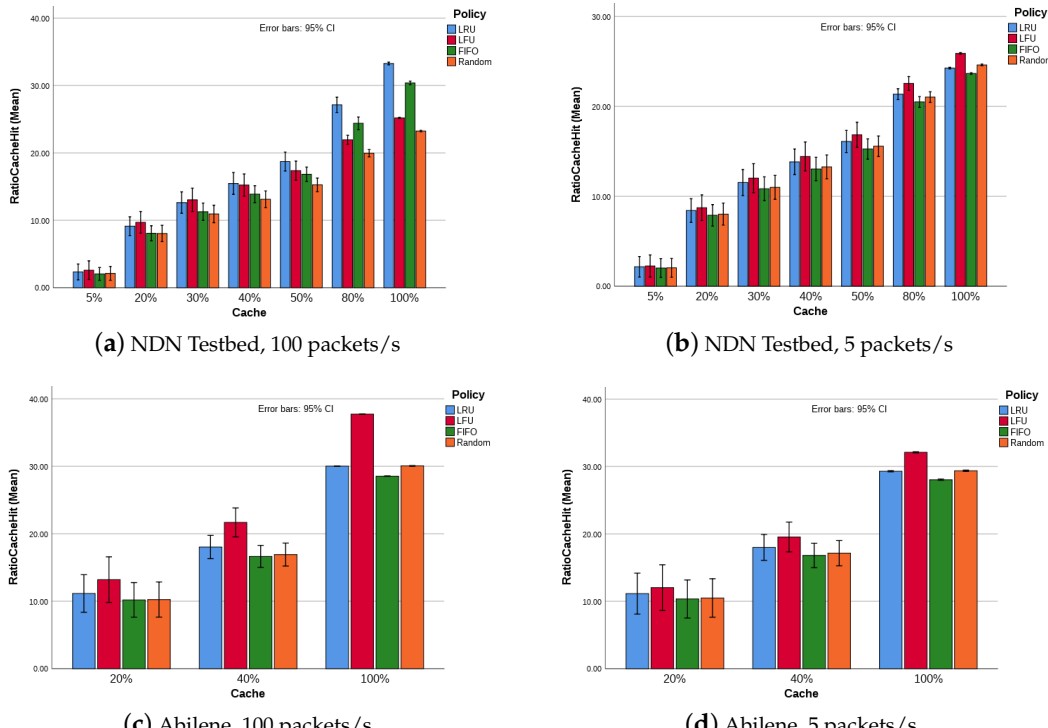

(**a**) NDN Testbed, 100 packets/s

(**b**) NDN Testbed, 5 packets/s

(**c**) Abilene, 100 packets/s

(**d**) Abilene, 5 packets/s

**Figure 2.** CHR evolution with variable number of CS, for scenarios with 41 consumers and 1 producer.

If we isolate a specific node/router, to which the consumer(s) is connected, we have for Topology 1 a particular node requesting 100 different contents from the same consumer, thus, in a given time period, the contents will be less popular than for Topology 2, where with a lesser request frequency (5 packets/s each by four or five consumers connected to a node), the number of new packets per equivalent time period is reduced. Therefore, as the results show, in Topology 1, LRU performs better than LFU. However, in Topology 2, LFU performs better because the contents become popular more easily.

A particular situation is presented on Topology 1, where LRU performs better for a consumer request rate of 100 packets/s and, LFU performs better when consumers request content at a rate of 5 packets/s. The justification is similar: with fewer requests for different content in a specific time period, the existing content stays in CS for longer. Combining the dimensions of LFU, time and access frequency, this policy will perform better.

The conclusion presented in the last paragraphs are further reinforced by the results from the aforementioned scenario with a more complex traffic (i.e., 32 consumers and 10 producers), as shown by Figure 3. With an increased traffic, LFU (combining time and access frequency) performs better. Figure 3c,d show the same simulation and behavior as the results shown in Figure 3a,b, respectively, but with $\alpha = 0.75$. As presented earlier in Section 3.1, the skewness factor affects the CHR, reducing it when $\alpha$ is low. However, as shown by the results, the relative behavior among the scenarios is not affected.

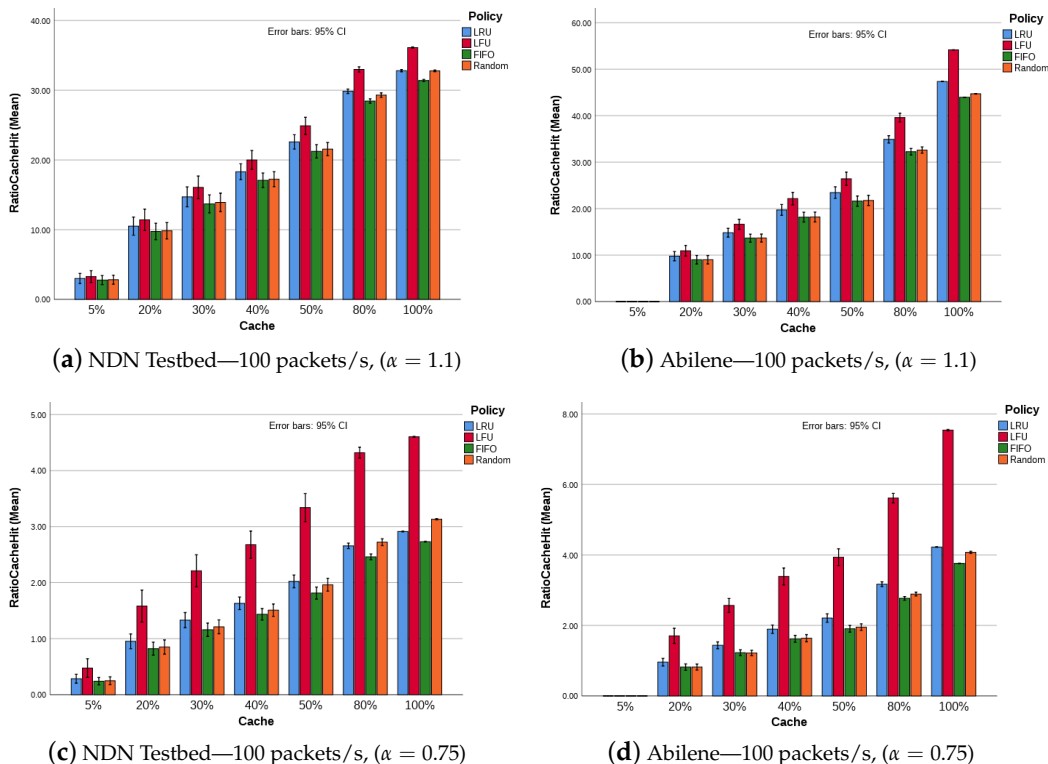

(**a**) NDN Testbed—100 packets/s, ($\alpha = 1.1$)

(**b**) Abilene—100 packets/s, ($\alpha = 1.1$)

(**c**) NDN Testbed—100 packets/s, ($\alpha = 0.75$)

(**d**) Abilene—100 packets/s, ($\alpha = 0.75$)

**Figure 3.** Cache hit ratio evolution with variable number of CS, for scenarios with 32 consumers and 10 producers.

LFU and random present lower sensitivity to the variation of CS than LRU and FIFO in a less complex and less busy network; see Figure 4a. This situation changes in a more complex network as shown in Figure 4b.

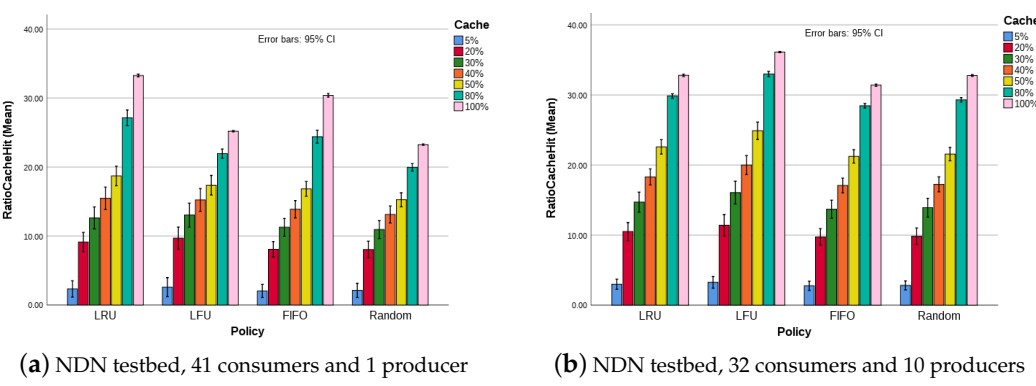

(**a**) NDN testbed, 41 consumers and 1 producer

(**b**) NDN testbed, 32 consumers and 10 producers

**Figure 4.** Sensibility of policies with variable number of CS, NDN testbed with content request rate of 100 packets/s.

Increasing network traffic affects the CHR. Limitation in terms of the available hardware restrained the simulation of all scenarios with a higher request rate. One scenario (i.e., Topology 1, with 100% cache nodes) was selected and simulated with a request rate of 1000 packets/s. Figure 5 shows the result of the aforementioned simulation, which is similar to the result achieved earlier and presented in Figures 2b and 3.

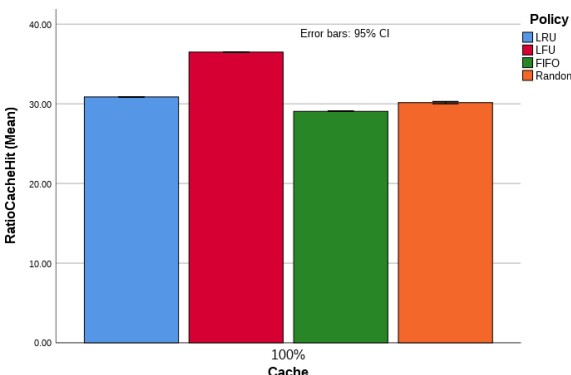

**Figure 5.** Cache hit ratio for NDN testbed with all routers equipped with CS and a content request rate of 1000 packets/s.

The results presented in this section are particularly inconsistent with those achieved in [5], where the network performance in terms of CHR increases when the number of CS increases up to about 40% and then decreases with the number of CS above this threshold. However, the results are consistent with ones achieved in [6], where the CHR increases with the continuous increasing in the number of CS, or in [7–16,50] which evaluate the network performance with a constant number of CS but different sizes.

### 5.2.2. Number of Upstream Hops

With more CS, the probability of finding the requested content closer to consumer is high, the long routes (or high number of hops) are avoided. This situation is shown in Figure 6a. Once again, this is true and verified for scenarios simulated with a high skewness factor, i.e., $\alpha = 1.1$. For $\alpha = 0.75$, as already verified for interest re-transmission and retrieval delay, this metric presents a relatively constant value; see Figure 6b.

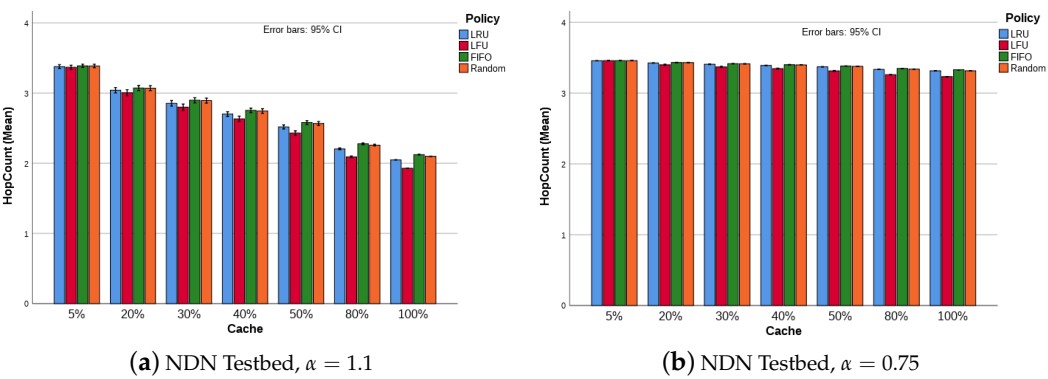

(**a**) NDN Testbed, $\alpha = 1.1$                                         (**b**) NDN Testbed, $\alpha = 0.75$

**Figure 6.** Upstream hops to content retrieval, NDN testbed with 32 consumers and 10 producers.

### 5.2.3. Network Traffic

Regarding the network behavior in terms of its traffic (i.e., number of interest/data messages), Figure 7 presents the obtained results, which, as expected, show that with a smaller number of CS, the network traffic is high and vice versa. The figure shows the result for a scenario with 32 consumers and 10 producers, which is similar to the result achieved with the scenario with 41 consumers and 1 producer. This behavior is justified by the following:

- For a scenario with less CS in the network, the interest will take a longer route before finding the requested content. In this process, the probability of delay (as shown in Figure 8, Section 5.2.4) and the possible discarding of the packet is high;
- With more CS, as noted earlier, the CHR will be high. That is, the probability of finding the requested content closer to the consumer is also high, avoiding a longer route (or

high number of hops, as explained in Section 5.2.2), and decreasing the probability of packet loss.

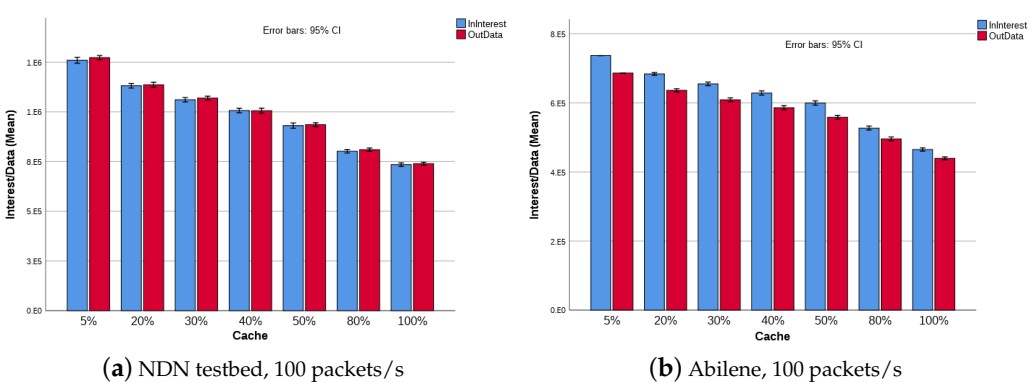

(**a**) NDN testbed, 100 packets/s          (**b**) Abilene, 100 packets/s

**Figure 7.** Network traffic, NDN testbed with 32 consumers and 10 producers, $\alpha = 1.1$.

Figure 7a shows the variation in traffic with the variation in CS in the network. This result is particularly different from the one achieved when the skewness factor is $\alpha = 0.75$, which presents constant traffic for all levels of cache (i.e., 5%–100%), for both Abilene and NDN testbed.

### 5.2.4. Retrieval Delay

Figure 8 shows the observed delay (in seconds) for each policy per scenario. This is the full delay, which includes the delay observed with re-transmissions (when they are observed).

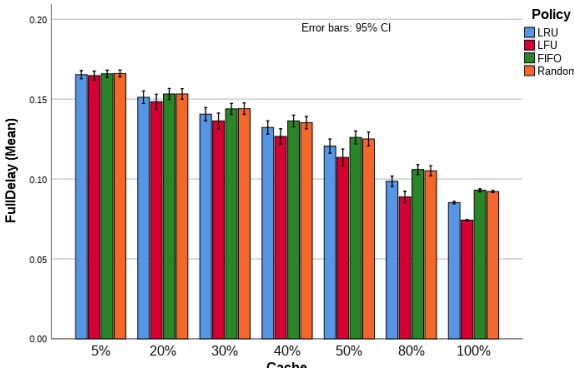

**Figure 8.** Full delay content fetch, NDN testbed with content 32 consumers and 10 producers, $\alpha = 1.1$.

With less CS, we have high delay because in this case, the interest is forwarded to more distant nodes, through more hops (as is explained in Section 5.2.2). That is, the response for a given interest is longer because the interest is sent through more nodes (hops) until it retrieves the content. When the number of CS is high, the probability of reaching the content provider with less hops is high. However, in this case, a relatively increased number of retransmissions is necessary when a content is not retrievable nearby and the upstream nodes send back a NACK. The result presented here is similar for both the topologies and both content request ratios. This metric presents a relatively constant value for scenarios with $\alpha = 0.75$, which is due to the absence of retransmission, as presented in the next section.

Although the results from [5] are inconsistent with the results presented in Section 5.2.1, related to CHR, they are consistent with the results obtained in this section for retrieval delay.

### 5.2.5. Interest Retransmissions

The consumer request rate and the consequent network traffic considered in this work is relatively low when compared to real networks. However, given the reduced size of the transmission buffer in each router considered for this work, some scenarios (i.e., NDN Testbed, and Abilene, with $\alpha = 1.1$) observed congestion and a consequent packet drop, triggering the NACK response and the interest retransmission from the consumers (see Figure 9a), which shows the number of retransmissions slightly increasing with the increased number of CS. However, scenarios with $\alpha = 0.75$ did not present retransmissions (see Figure 9b).

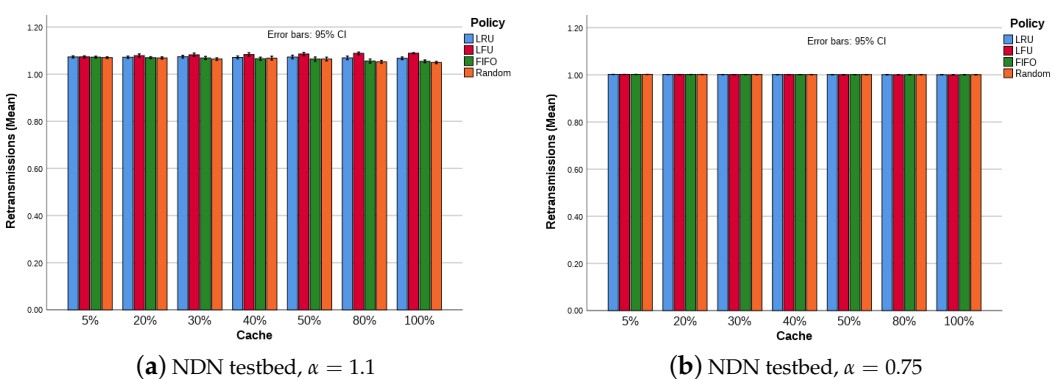

(**a**) NDN testbed, $\alpha = 1.1$      (**b**) NDN testbed, $\alpha = 0.75$

**Figure 9.** Interest re-transmissions, NDN testbed with 32 consumers and 10 producers.

## 6. Conclusions

This work evaluated the performance of four caching policies (i.e., LRU, LFU, FIFO and random) in two network topologies (i.e., NDN testbed and Abilene) configured with seven different scenarios (different number of CS), and with two different content request rates, 100 and 5 packets/s. Additionally, one specific scenario with a consumer request rate of 1000 packets/s was simulated. For each scenario, two configurations were considered: one with 41 consumers and 1 producer and another composed of 32 consumers and 10 producers. The requests were considered, following the MZipf distribution. Two values of the skewness factor were chosen in order to evaluate the performance considering two different content types. The results were achieved through simulations using ndnSIM.

In general, LFU performs better for all considered metrics. For NDN testbed, random presents similar performance as the LRU replacement policy when the network is equipped with CS in more than 80% of its routers. A different situation is presented by the Abilene network, where LRU presents a slightly higher performance than random.

For a lower value of $\alpha$, the popularity of contents in the MZipf distribution decreases, thus decreasing the CHR. For all scenarios, the retrieval delay, traffic, and hop-count present relatively constant but high values compared to the configuration with higher $\alpha$. This is expected because with lower popularity, the interest takes more time and goes through a longer route to fetch the requested content.

Although desirable, the evaluation of the performance with a higher content request rate was limited by the available computational resources. This limitation may be overcome in future, and this work may then be extended. Moreover, as referred to earlier, the extension of this work to include more caching replacement policies, combining different caching decision strategies, may be considered. Additionally, for future work, it would be interesting to evaluate the performance of these policies in a dynamic network, such as the *vehicular ad hoc network* (VANET), similar to the work in [64], which is specific to caching decision strategies.

**Author Contributions:** Conceptualization, E.T.d.S. and J.M.H.d.M.; methodology, E.T.d.S.; software, E.T.d.S.; validation, E.T.d.S., J.M.H.d.M. and A.L.D.C.; formal analysis, E.T.d.S.; investigation, E.T.d.S.; resources, E.T.d.S., J.M.H.d.M. and A.L.D.C.; data curation, E.T.d.S.; writing—original draft preparation, E.T.d.S.; writing—review and editing, E.T.d.S., J.M.H.d.M. and A.L.D.C.; visualization, E.T.d.S.; supervision, E.T.d.S., J.M.H.d.M. and A.L.D.C.; funding acquisition, J.M.H.d.M. and A.L.D.C. All authors have read and agreed to the published version of the manuscript.

**Funding:** This work has been supported by FCT – Fundação para a Ciência e Tecnologia within the R&D Units Project Scope: UIDB/00319/2020.

**Institutional Review Board Statement:** Not applicable.

**Informed Consent Statement:** Not applicable.

**Data Availability Statement:** The data presented in this study are openly available in zenodo.org accessed on 24 January 2022, at https://doi.org/10.5281/zenodo.5902395 accessed on 24 January 2022.

**Conflicts of Interest:** The authors declare no conflict of interest.

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
