# Peer review of "NDN Content Store and Caching Policies: Performance Evaluation"

_computers, doi:10.3390/computers11030037_

Round 1
Reviewer 1 Report
The paper presents a comprehensive analysis of ndsSIM simulation on caching performance based on different scenarios. It was well written and can be used as guidance and reference when applying those caching strategies and policies based on individual use cases. I recommend the paper accepted with minor English and style corrections.
Reviewer 2 Report
In this manuscript, the authors aim at evaluating the performance of caching replacement policies in a network topology with varying multiple parameters. They do so through simulation using an NDN testbed and compare their results with the literature. Overall, the manuscript is well-written and easy to follow and understand. However, I have the following comments:
- A language proof-reading is required. Some examples of grammar or style mistakes:
- Line 25, the indefinite article “a” is missing from the sentence.
- Lines 51 to 54. Style mistake, the use of “1) complementing”, then “2) extend” and “3) highlight”. Why not write “extending” and “highlighting”, to follow the same style?
- Line 166, missing space.
- It is not clear why the authors sometime refer to subsections using the word “Session”, as in line 48 and 50, while other times using the word “Section” as in line 300.
- Line 418, missing letter “s” in the word “configuration”.
- Out of 54 references, only 5 are from the last 5 years (2017-2022). There needs to be more recent references.
Strengths: Well-written, good comparisons and commenting.
Weaknesses: Use of non-recent state-of-the art and not much novelty in contribution.
Reviewer 3 Report
There is continued interest in the manuscript titled "NDN Content Store and Caching Policies: Performance Evaluation". There are four comments:
1. The whole article is clear and rigorous, and the conclusion is convincing.
2. This article evaluates the performance of four caching strategies and Content Store in different scenarios through four indicators in NDN, and compares them to obtain the corresponding experimental results, which has certain research significance.
3. The innovation points are clear, and it has certain reference for future research.
4. The content of this article is substantial and in line with the real scene, but the contrast between the data in some figures cannot be clearly seen.
